# A Simple Method for Preparation of Highly Conductive Nitrogen/Phosphorus-Doped Carbon Nanofiber Films

**DOI:** 10.3390/ma15175955

**Published:** 2022-08-29

**Authors:** Tongzhou Chen, Yongbo Chi, Xingyao Liu, Xiwen Xia, Yousi Chen, Jian Xu, Yujie Song

**Affiliations:** 1State Key Laboratory of Fine Chemicals, Frontiers Science Center for Smart Materials Oriented Chemical Engineering, School of Chemical Engineering, Dalian University of Technology, Dalian 116024, China; 2Engineering Laboratory of Advanced Energy Materials, Ningbo Institute of Material Technology and Engineering, Chinese Academy of Sciences, Ningbo 315201, China; 3School of Materials Science and Chemical Engineering, Ningbo University, Ningbo 315201, China

**Keywords:** carbon fiber films, heterogeneous elements, electrospinning, electrical conductivity

## Abstract

Heteroatom-doped conductive carbon nanomaterials are promising for energy and catalysis applications, but there are few reports on increasing their heteroatom doping content and conductivity simultaneously. In this manuscript, we use 2-(4-aminophenyl)-5-aminobenzimidazole as the diamine monomer to prepare polyamic acid with asymmetric structural units doped with phosphoric acid (PA) and polyacrylonitrile (PAN) as innovative composite precursors, which are then electrospun into nanofiber films. After stabilization and carbonization, the electrospun fibers are converted into N/P co-doped electrospun carbon nanofiber films (ECNFs) with high heteroatom content, including 4.33% N and 0.98% P. The morphology, structure, and conductivity of ECNFs were systematically characterized. The ECNFs doped with 15 wt.% PA exhibited conductivity that was 47.3% higher than that of the ECNFs undoped with PA, but the BET surface area decreased by 23%. The doped PA in the precursor nanofibers participated in the cyclization of PAN during thermal stabilization, as indicated by infrared spectroscopy and thermogravimetric analysis results. X-ray diffraction and Raman results indicate that a moderate amount of PA doping facilitated the formation of ordered graphitic crystallite structures during carbonization and improved the conductivity of ECNFs.

## 1. Introduction

Energy demand continuously increases along with rapid global population growth. Uncontrolled development will cause serious environmental pollution and threaten the livelihood of human beings because nonrenewable energy resources are limited. Developing an environmentally friendly, efficient, and recyclable new energy source is imperative to solve the energy crisis and achieve sustainable development. At present, the most promising alternatives for the capacity to efficiently convert and store renewable energy are fuel cells [1,2] and metal-air batteries [3]. However, the sluggish kinetics of the oxygen reduction reaction (ORR) occurring at the cathode obstructs energy conversion efficiency. To date, Pt and its alloys are the best catalysts for ORR [4]. Pt-based catalysts suffer from scarcity, high cost, and easy poisoning, which has seriously hampered the large-scale applications of these promising energy systems.

Heteroatom-doped carbon nanomaterials including graphene [5], carbon nanotubes [6], carbon nanofibers [7], mesoporous graphitic arrays [8], and porous carbon [9] are considered some of the most promising alternatives to Pt-based catalysts due to their durability and cost-effectiveness. Additionally, these materials do not suffer CO poisoning or methanol crossover effects. Many heteroatom-doped carbon nanomaterials, including those doped with N [10], P [11], B [12], S [13], and F [14], possess excellent performance for ORR. Furthermore, co-doping with heteroatoms can produce more defects due to the synergistic effect between them, and these catalysts exhibit even higher electrocatalytic activities for ORR than their single-atom-doped counterparts [15]. Extensive research has been focused on the effect of doping with different elements’ content on ORR activity. However, the conductivity of the electrode material is also crucial in ORR because the effective potential acting at the catalytic site depends on the electrical conductivity. Increasing doping content and conductivity of carbon materials simultaneously is regarded as an underexplored problem, as conductivity is generally inversely proportional to doping content [16,17]. Thus, it is essential to explore novel metal-free carbon materials with high conductivity and high doping content for electrode materials in ORR.

Electrospun carbon nanofiber films (ECNFs) are a class of carbon nanomaterials with many potential applications, which are prepared through electrospinning. The electrospinning process is fast, simple, and relatively inexpensive, and it can also produce fibers from very small amounts of polymer, making it ideal for evaluating novel polymers in fiber form. Due to the diameter of electrospun fibers on the nanometer scale and excellent uniformity, ECNFs produced by this technology have high porosities and high specific surface area [18]. These characteristics make ECNFs produced by electrospinning attractive for many applications, especially as electrode materials for supercapacitors and fuel cells. The main raw material of ECNFs is polyacrylonitrile (PAN) [19]. However, PAN-based ECNFs have low electrical conductivity due to the helical conformation of their molecules, making them difficult to graphitize. In contrast, polyimide (PI) has a rigid chain with high chemical regularity and is easier to graphitize than PAN [20,21]. The conductivity of PI-based ECNFs after carbonization at 1000 °C is 2.5 S/cm, which is twice that of PAN-based ECNFs under the same conditions [22]. Therefore, PI-based ECNFs have great potential due to their high conductivity and high doping content for ECNFs.

The influencing factors of conductivity and doping content of PI-based ECNFs include intrinsic material structure and material modification. In terms of intrinsic material structure, the molecular structure of dianhydride and diamine monomers is the key factor affecting the carbonization behavior and conductivity of PI-based ECNFs [21]. Aromatic heterocycles can be introduced to increase the content of heteroatoms after carbonization at the same time [8]. In another material modification, introducing mesophase pitch [23], carbon nanotubes [24], PAN [25], and boric acid [26,27] into ECNF precursors for blending modification is also an effective method to improve the conductivity and increase the doping content of ECNFs. Among them, PAN and polyamic acid (PAA) have good compatibility [25,28] and improve the spinnability of PAA [25].

In this manuscript, we introduce an asymmetric structural benzimidazole ring into the PI system to provide excellent electrical properties and high N content after carbonization. Blending PAN into the system improves the spinnability of polymer solution while increasing the conductivity of carbonized nanofibers [25]. Finally, doping with phosphoric acid (PA) as a P source configures the spinning solution. Nanofiber films were prepared by electrospinning, and ECNFs were prepared by air thermal stabilization and N carbonization. The effect of PA content on the thermal stabilization, carbonization behavior, doping content, and conductivity of ECNFs was investigated, and the correlation between structure and properties was also investigated.

## 2. Experiment

### 2.1. Materials

3,3′,4,4′-Biphenyl tetracarboxylic dianhydride (BPDA), pyromellitic dianhydride (PMDA), PA, and N,N-dimethylacetamide (DMAc) were purchased from Shanghai Aladdin Biochemical Technology Co., Ltd. (Shanghai, China). 2-(4-aminophenyl)-5-aminobenzimidazole (BIA) was purchased from Anhui Senrise Technology Co., Ltd. (Fuyang, China). PAN (*M*_W_ = 90,000) was obtained from Jilin Chemical Fiber Co., Ltd. (Jilin, China). The materials were used without further purification.

### 2.2. Synthesis of PAA

The synthesis of PAA was accomplished using a typical synthetic route with some modifications. In a typical experiment, 4.46 mmol BIA was dispersed in DMAc and stirred for 30 min. Then, 0.4505 mmol BPDA was put into the solution in an ice-water bath and stirred for 4 h. Subsequently, 4.0545 mmol PMDA was slowly added into the solution in batches and stirred until the rotor of the overhead stirrer did not move. The obtained PAA solution was then stored in the refrigerator for 24 h to make the molecular weight uniform for further use.

### 2.3. Preparation of Spinning Solution

PAN powder was added into DMAc and stirred until completely dissolved. The mixed solution was prepared according to the 4:1 weight ratio of PAA:PAN. Next, PA was put into the PAA/PAN solution and stirred for 1 h for further use.

### 2.4. Preparation of Electrospun Nanofiber Films (ENFs)

The electrospinning process was carried out with a feed rate of 0.5 mL/h, a spinneret diameter of 0.5 mm, an applied voltage of 18 kV, and a distance of 18 cm between the spinneret and the collector. The humidity and temperature of the electrospinning chamber were 30 ± 5% and 30 ± 2 °C, respectively. After electrospinning, the as-spun nanofiber films were kept in a vacuum oven at 80 °C for 2 h. Throughout this manuscript, the obtained ENFs are referred to as ENFs-0, ENFs-15, and ENFs-30 according to the PA weight fractions of 0%, 15%, and 30%, respectively.

### 2.5. Preparation of ECNFs

For thermal stabilization, the ENFs were placed in a tube furnace in air for 1 h successively at 100 °C, 200 °C, and 300 °C, followed by 30 min at 350 °C. Then, the ENFs were carbonized for 1 h in Ar at 1000 °C and a ramp rate of 3 °C/min was used between the 100 °C, 200 °C, 300 °C, 350 °C, and 1000 °C plateaus. The obtained ECNFs are labeled as ECNFs-0, ECNFs-15, and ECNFs-30 throughout the manuscript.

### 2.6. Characterization

Thermal properties were evaluated using thermogravimetric analysis (TG, TGA 8000-Spectrum two-Clarus SQ8T) and a differential scanning calorimeter (DSC, DSC 214 thermal analyzer) at a heating rate of 10 °C/min. The molecular chain structures were analyzed by Fourier transform infrared spectrometry (FTIR, Cary660 + 620 infrared spectrometer). The surface morphologies of ECNFs were examined by scanning electron microscopy (SEM, S-4800). The surface elemental composition of each sample was detected by X-ray photoelectron spectroscopy (XPS, Thermo Scientific K-Alpha, Waltham, MA, USA). X-ray diffraction (XRD) patterns were obtained from an X-ray diffractometer (AXS D8 ADVANCE) with Cu Kα radiation (λ = 1.5406 Å) to investigate the crystal structure of each sample. The extent of graphitization of the ECNFs was evaluated using a Raman spectrometer (Renishaw inVia Reflex) with an excitation wavelength of 532 nm. An ASAP 2460 (Micromeritics, Norcross, GA, USA) was used to carry out N_2_ adsorption–desorption measurements. The specific surface area of ECNFs with different PA weight fractions was analyzed by the multipoint Brunauer–Emmett–Teller (BET) method.

### 2.7. Conductivity Measurements

The resistances (*R*) of the ECNFs were measured by a square resistance meter with a four-point probe method. The thicknesses (*t*) of the ECNFs were measured by a micrometer with a dial comparator. The conductivity (*σ*) of each sample was calculated using the following formula:$$\sigma =\frac{1}{Rt}$$

## 3. Results and Discussion

### 3.1. Characterization

Changes in the FTIR spectra of the ENFs before and after stabilization are shown in Figure 1. The characteristic absorption bands at 2242 cm^−1^, 1720 cm^−1^, and 1540 cm^−1^ were due to the C≡N stretching vibrations of PAN, C=O symmetric stretching vibration of the carboxyl group in amic acid, and C=O stretching vibration of the amide group, respectively [28,29]. The appearance of the band at 1600 cm^−1^ (C=N benzimidazole ring stretching vibration) [30] indicates that asymmetrically structured PAA was present in the ENFs. The very broad band complex from 2500 cm^−1^ to 3700 cm^−1^ is attributed to the overlap of the N-H stretching vibration of PAA and CH_2_ stretching vibrations of PAN [31]. When the precursor nanofibers were doped with PA, broad peaks shifting to lower wavenumbers should have been related to the protonation of the nitrogen of the imide by transferring one or more protons from PA to imidazole groups of the precursor nanofibers. Simultaneously, the band at 970 cm^−1^ (absorption band of monoprotonated phosphate HPO_4_^2−^) and 508 cm^−1^ (absorption band of diprotonated phosphate H_2_PO_4_^−^) appeared [30,32]. After the thermal treatment process, PAA was converted into the corresponding PI, and the linear PAN macromolecules were converted into aromatic ladder structures. For the PI moiety, the characteristic peaks at 1775 cm^−1^ (C=O of cyclic imide asymmetric stretching), 1722 cm^−1^ (C=O of cyclic imide symmetric stretching), 1360 cm^−1^ (C-N-C axial stretching), and 727 cm^−1^ (C=O flexural vibration) suggest the formation of cyclic imide rings [28,29]. For the PAN moiety, the C≡N absorption band of PAN disappeared and a band at 1602 cm^−1^ (C=N stretching vibration) appeared. These changes in the spectra indicate that the PAN molecules were converted to a ladder structure. In addition, the intensity of the diprotonated phosphate H_2_PO_4_^−^ absorption band of ENF-15 weakened, and the monoprotonated phosphate HPO_4_^2−^ absorption band of ENFs-15 disappeared. These differences are attributed to the PA participating in the cyclization and aromatization of PAN [33], and the excess PA still existed in the fiber as protons.

Figure 2a and Table 1 display the DSC curves and peak data of ENFs in air. The DSC curve possessed two endothermic peaks, most likely due to solvent evaporation and imidization. There was also an exothermic peak due to the stabilization of PAN. The endothermic peak of ENFs-30 was 10 °C lower than that of ENFs-15 and 30 °C lower than that of ENFs-0, which was due to the protonated PA that promoted the dehydration cyclization of PAA [34]. The T_i_ and T_p,C_ of the cyclization reaction moved to lower temperature, and the peak shape flattened with increasing PA weight fraction. These results suggest that PA reduced the activation energy of the cyclization reaction of PAN precursor and moderated the concentrated heat release. The DSC curve shows that the composite precursor underwent complete imidization and cyclization and that it reached a thermally stabilized state after heat treatment at 350 °C in air.

Figure 2b displays the thermogravimetric behavior of ENFs in air and N_2_ atmospheres. The materials lost weight in five steps: 50–100 °C, 130–230 °C, 290–350 °C, 530–650 °C, and 750–850 °C. These five steps can be assigned to solvent evaporation, imidization, dehydrogenation reactions of PAN and PA, carbonization, and the decomposition of the phosphate group [35].

The DSC and TGA results indicate that the doped PA promoted the imidization of PAA and the cyclization of PAN, leading to a higher quality of the fibers during the stabilization process in the air. The molecular structures of precursors during the stabilization in air are shown in Figure 3. The higher quality of the fibers containing PA during the pyrolysis of the thermally stabilized structure indicates that PA effectively enhanced the formation of ordered graphitic structures. However, the quality of the fibers was reduced after 1000 °C carbonization due to the decomposition of the phosphate group at 810 °C. According to the TG, the total carbon yield of ENFs-30 was 46.1%, which is 11% lower than that of ENFs-0 and 8% lower than that of ENFs-15. Because the heating rate of thermal analysis was 10 °C/min, which was faster than the actual carbonization, more degradation side reactions could occur, so the actual total carbon yield would be higher than what is indicated by the thermogravimetric test results [20].

The surface morphologies of the ECNFs made from the precursors of nanofiber films with the PA weight fractions of 0%, 15%, and 30% are shown in Figure 4. The surface of all single fibers was dense and void-free. As a result of bending instability [36], the single fiber arrangement in the obtained ECNFs was random and chaotic. Additionally, not all of the ECNFs contained beads or beaded nanofibers. When the doping content of PA was increased to 15% and 30%, the diameters of each single fiber increased from 400 nm to 420 nm and 500 nm, respectively. The ECNFs made from the precursor nanofibers with PA showed greater thermal weight loss, whereas the obtained ECNFs were thicker, suggesting that the majority of PA in the spinning solution existed as protons instead of ions [33]. Due to the hydrogen bonding in PA, the PA-doped spinning solution was more viscous, and so the fibers obtained were thicker from spinning under the same conditions.

XPS was performed to investigate the elemental compositions of the ECNFs. Figure 5a shows that the elements of C, O, and N existed in all ECNFs. Furthermore, the XPS survey spectra of ECNFs-15 and ECNFs-30 exhibited an obvious P 2p peak, confirming successful co-doping with N and P. The corresponding atomic percentages are listed in Table 2. The P contents of ECNFs-15 and ECNFs-30 were the same, indicating that the PA that reacted with PAN could partially remain in ECNFs, and the excess protonated phosphate was completely removed after heat treatment. In addition, with increasing PA weight fractions, the N content of the corresponding carbonized fibers increased, which may be because the PA group decomposed at 810 °C. During decomposition, the PA group emitted P, which led to a decreased rate of N removal. The P and N contents of PA-doped ECNFs were simultaneously enhanced, and correspondingly the carbon content was slightly decreased. As shown in Figure 5b, the N 1s spectrum could be deconvoluted into three components: The peaks at 398.2, 400.8, and 402.4 eV were ascribed to pyridinic N, graphitic N, and pyridinic-N–oxide [37,38], respectively. As shown in Figure 5c, the high-resolution P 2p XPS peaks located at ca. 129.6, 132, 134.2, and 136 eV could be attributed to elemental phosphorus, P–C bonds, P–O bonds, and phosphorus oxide P_2_O_5_ [39,40], respectively. The components were not all present in each ECNF. According to the deconvolution results in Table 3, when the PA weight fraction increased the N-O bond and N=C bond in the pyridine-N–oxide were transformed into graphitic N, and the excess oxygen formed a P-O bond and P=O bond. Pyridinic N and graphitic N were critical for forming active sites for ORR [41,42,43], and the introduction of P generated new active sites for ORR [38].

Figure 6 shows the X-ray diffraction patterns of the ECNFs. The two diffraction peaks centered at 2θ angles of 24° and 43° were attributed to the (002) and (100) crystallographic planes of graphite, respectively [44]. The 2θ and the full-width-at-half-maximum value obtained from the (002) peak were used to calculate planar spacing d_002_ using the Bragg equation, and the crystallite size L_c_ and the mean stack layers L_c_/d_002_ were calculated with the Scherrer equation, and the results are shown in Table 4. The values of d_002_ increased from 0.354 nm to 0.365 nm after PA was doped in the precursor nanofibers. Meanwhile, no appreciable change in the d_002_ value was observed with further increases in PA weight fractions. Additionally, the values of L_c_ and L_c_/d_002_ became significantly larger after PA was doped in the precursor nanofibers. When the PA weight fraction increased, the values of L_c_ and L_c_/d_002_ decreased. These results suggest that an appropriate amount of PA was an effective promoter for the formation of ordered graphitic crystallite structures in ECNFs, whereas the structure of ECNFs became more disordered due to the insertion of excess P atoms. Furthermore, a magnified inset of the spectra surrounding the (100) plane shows that the characteristic peak of the graphite microcrystalline structure belonging to the (100) plane in ECNFs-15 and ECNFs-30 appeared, indicating that the doping of PA is beneficial to the growth of the width of the microcrystalline basal plane.

Raman spectroscopy was used to further study the structural characteristics of the as-prepared samples. Figure 7 shows the Raman spectra of ECNFs. The Raman spectra of carbonaceous materials possessed two characteristic bands. The first band was the “D-band,” centered at a Raman shift of 1350 cm^−1^ and produced by an imperfection or lack of hexagonal symmetry at the edges of graphitic layers. The second band was the “G-band,” centered at a Raman shift of 1590 cm^−1^. This peak corresponds to an ideal graphitic lattice vibrational mode with E_2g_ symmetry relative to the motion of sp^2^-bonded carbon atoms [45]. The integral area values of the D and G bands (I_D_/I_G_, known as the “R-value”) were calculated to characterize the graphitization degree. The corresponding R-values of the ECNFs decreased from 3.24 to 2.76 as the PA weight fraction increased from 0% to 15%. This result is further indication that the presence of PA promoted the formation of ordered graphitic crystallite structures in the ECNFs. Simultaneously, the R-values of the ECNFs decreased to 2.94 when the PA weight fraction increased to 30%, suggesting an increase in defects in the graphitic structure due to the insertion of excess P atoms. This observation is consistent with the XRD results. The P content of ECNFs-30 did not increase, indicating that P atoms were excluded from the fibers during carbonization, leaving vacancy defects.

Figure 8 exhibits N_2_ adsorption–desorption isotherms of the ECNFs, which indicates the obvious characterization of type II isotherms. According to the isotherms, S_BET_ of the ECNFs were obtained, and the results are shown in Table 4. Results show that ECNFs-15 had the lowest BET surface area due to the highest degree of graphitization, resulting in a dense fiber surface. Their bigger single-fiber diameter also led to the lower BET surface area. However, ECNFs-30 had the biggest fiber diameter and the lowest degree of graphitization, which were obtained with the highest BET surface area. this may be due to the activation of excess PA on the surface when carbon nanofiber molding formed the rough surface, resulting in an increase in the BET surface area. Unusually, the desorption curves of the P-doped ECNFs are below the adsorption curves. The reason is speculated as being that the separate phosphorus phase was in the voids between carbon crystallites [40], since easily accessible elemental phosphorus cannot survive in ultra-high vacuum, which is necessary for N_2_ adsorption–desorption measurements.

### 3.2. Conductivity Analysis

From the conductivity values (σ) of ECNFs in Figure 9, it can be seen that the σ of ECNFs without PA was 10.72 S·cm^−1^, which is significantly higher than the 2.5 S·cm^−1^ of ECNFs of the BPDA and p-phenylenediamine system [20,46], indicating that the introduction of a BIA asymmetric structural unit into the PAA system and the simultaneous doping of PAN improved electrical conductivity. After doping with PA, the σ of ECNFs was further improved to 15.79 S·cm^−1^. As the PA weight fraction increased, increasing defects in the graphitic structure led to a decrease in electrical conductivity, but it was still higher than that of undoped PA ECNFs. The increase in the σ may have been due to the close contact and interwoven microstructures constructed by the different types of precursors during the carbonization of the PAA/PAN composites. Furthermore, the R-value was inversely proportional to the conductivity value, meaning that the high carbonization degree of fibers greatly improved the σ of the ECNFs.

Due to the bending instability of electrospinning, the nanofibers in the ECNFs were randomly and chaotically arranged. Therefore, the conductivity of ECNFs depended not only on the conductivity of individual nanofibers but also on the contact resistance at the contact locations between nanofibers. In this study, all of the ECNFs possessed similar morphologies, and likely there were similar contact resistances between nanofibers. The conductivity differences of ECNFs were therefore attributed to the different graphitic crystallite structures that resulted from the different weight fractions of PA in the precursor nanofibers. To sum up the above-acquired results, the PA weight fraction of 15% in ECNFs was optimal with an N content of 4.33%, a P content of 0.98%, and a σ of 15.79 S/cm. It is worth noting that the resistance values obtained from the measurements of ECNFs are composed of the resistance values of individual nanofiber and contact resistance. Therefore, the conductivity of individual nanofibers may have been higher than that of the ECNFs.

## 4. Conclusions

In this study, ECNFs co-doped with N and P were simply synthesized by electrospinning precursor solutions and subsequent thermal treatments. ECNFs doped with PA showed more moderate pyrolysis behavior before 800 °C, but the thermal stability decreased after PA decomposition, and the weight retention rate of ECNFs-15 after carbonization at 1000 °C was 49.7%. The doped PA participated in the PAN cyclization reaction during stabilization, leading to a P-atom content of ECNFs doped with PA of 0.98%. After carbonization, heteroelement defects were formed, d_002_ increased, and simultaneously the PA group promoted the formation of an ordered graphite crystallite structure so that the crystallite size L_c_ of ECNFs-15 was 1.85 nm, the mean stack layer L_c_/d_002_ value was 5.07, and the R-value was 2.76, which was improved compared to that of undoped PA ECNTs. The asymmetric structural unit of BIA in PAA was entangled with the PAN molecular chain in the helical conformation, which was closely intertwined during carbonization, which was beneficial for improved electrical conductivity.

Asymmetric diamine and dianhydride were used as raw materials to prepare PAA, and an appropriate amount of PAN and PA was introduced to obtain ECNFs with an N content of 4.33%, a P content of 0.98%, and a σ of 15.79 S/cm, which was significantly higher than that of ECNFs prepared from other precursors. Taken together, the methods described in this manuscript provide a synthesis route for materials with high heteroelement content and highly tunable conductivity.

## Figures and Tables

**Figure 1 materials-15-05955-f001:**
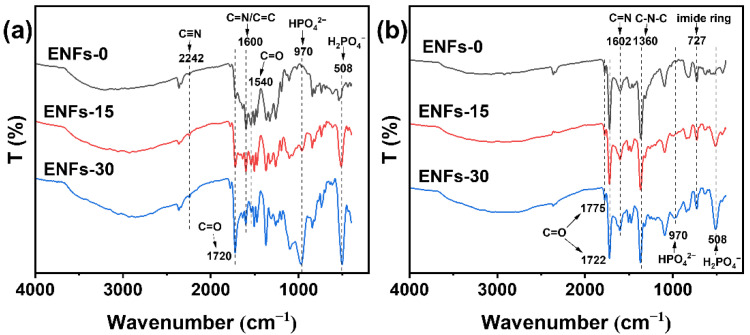
FTIR of ENFs with different PA weight fractions before (**a**) and after (**b**) heat stabilization.

**Figure 2 materials-15-05955-f002:**
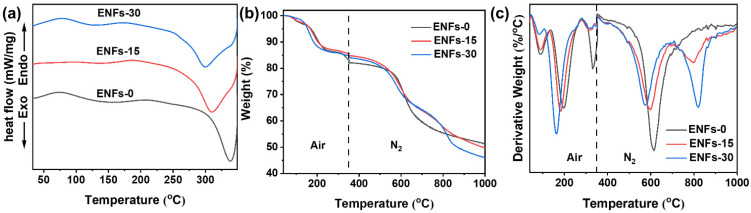
(**a**) DSC curves of ENFs with different PA weight fractions under an atmosphere of air. (**b**) TG and (**c**) DTG curves of ENFs with different PA weight fractions under air and N_2_ atmospheres.

**Figure 3 materials-15-05955-f003:**
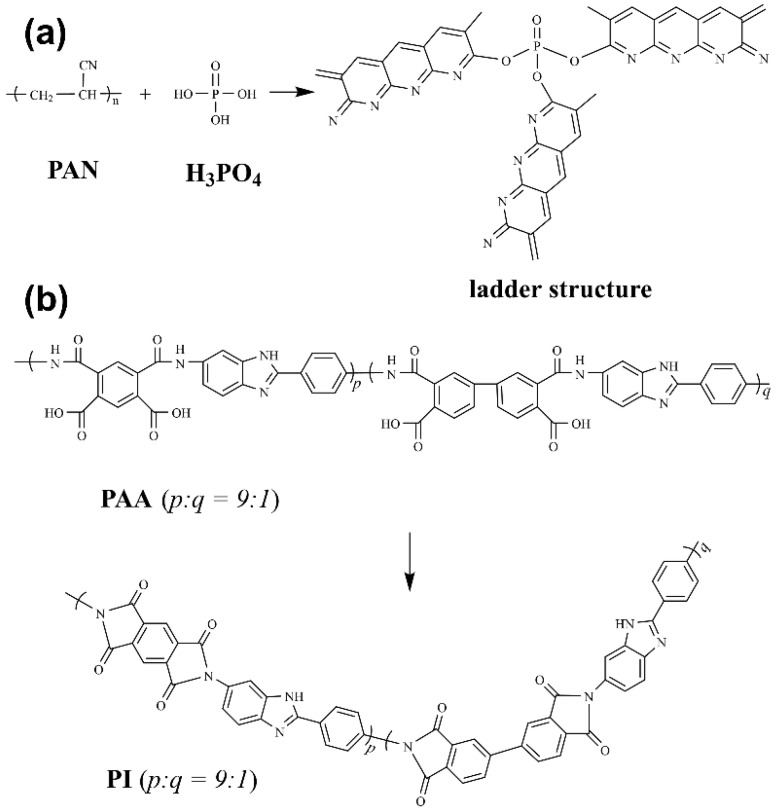
Molecular structures of (**a**) PAN and (**b**) PI before and after stabilization in air.

**Figure 4 materials-15-05955-f004:**
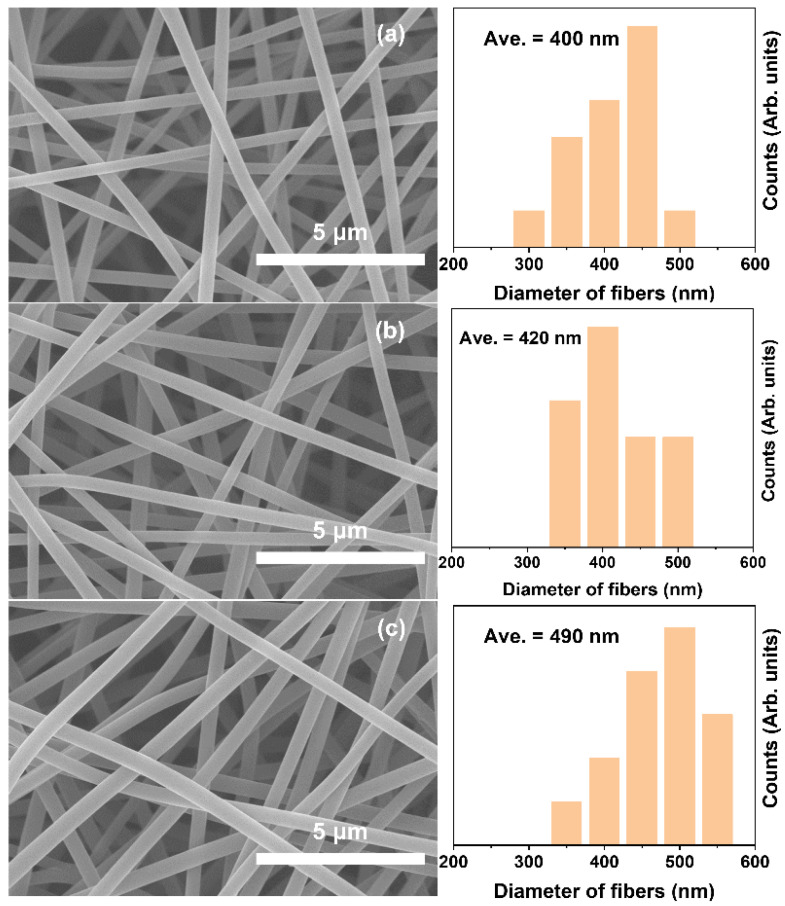
SEM images and the corresponding diameter distributions of ECNFs with different PA weight fractions: (**a**) ECNFs-0, (**b**) ECNFs-15, (**c**) ECNFs-30.

**Figure 5 materials-15-05955-f005:**
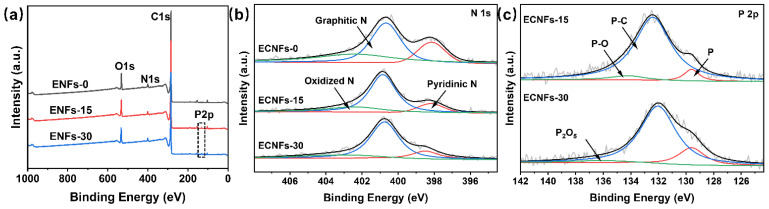
The wide XPS survey spectra (**a**), high-resolution N 1s XPS spectra (**b**), and high-resolution P 2p XPS spectra (**c**) of ECNFs with different PA weight fractions.

**Figure 6 materials-15-05955-f006:**
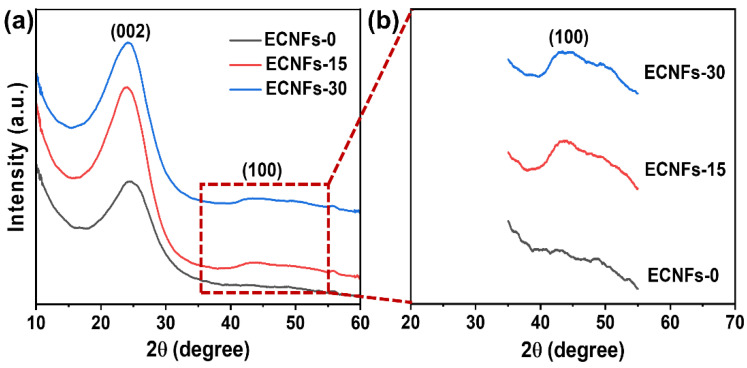
XRD spectra of ECNFs with different PA weight fractions (**a**) and the magnified inset of the (100) peak (**b**).

**Figure 7 materials-15-05955-f007:**
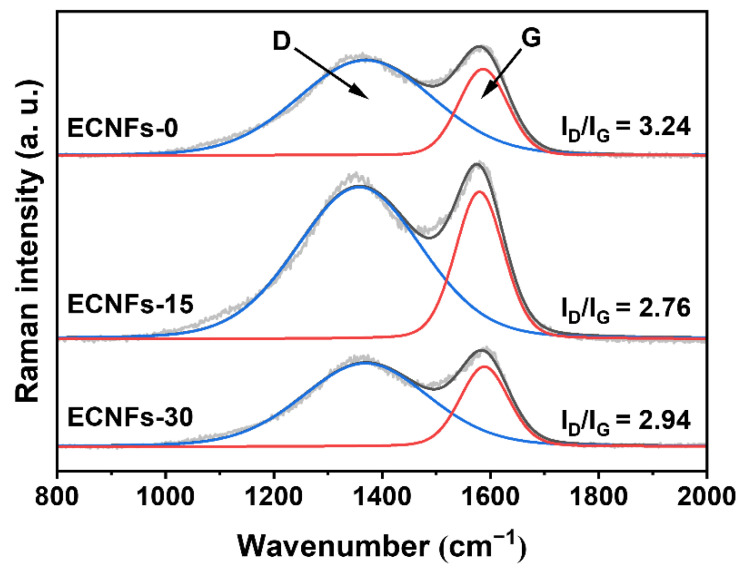
Deconvoluted Raman spectra and the corresponding R-value of ECNFs with different PA weight fractions.

**Figure 8 materials-15-05955-f008:**
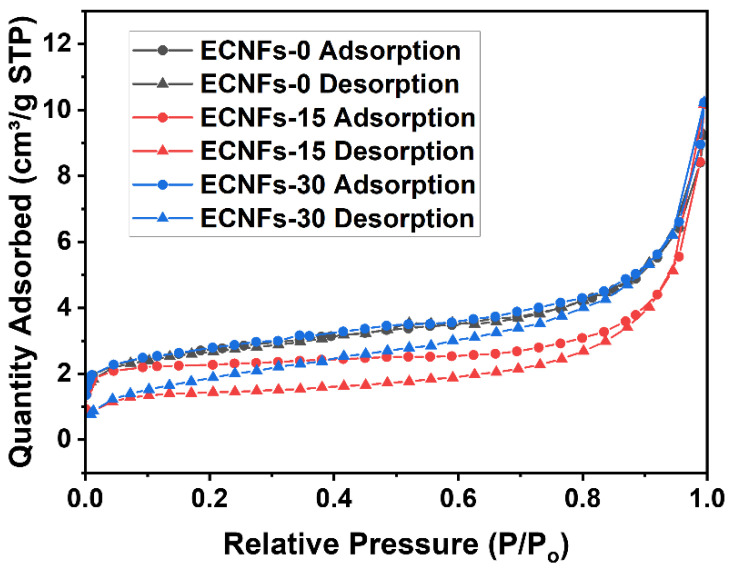
N_2_ adsorption–desorption isotherms of ECNFs with different PA weight fractions.

**Figure 9 materials-15-05955-f009:**
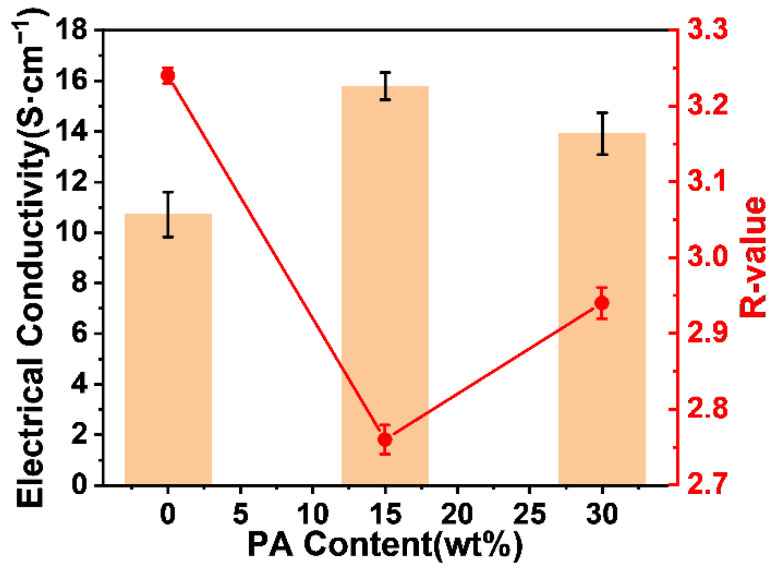
Electrical conductivity and the corresponding R-value of ECNFs with different PA weight fractions.

**Table 1 materials-15-05955-t001:** DSC data of ENFs with different PA weight fractions.

	T_p,I_/°C (Imidization)	T_i__,C_/°C (Cyclization)	T_p,C_/°C (Cyclization)
ENFs-0	209	309	339
ENFs-15	187	278	310
ENFs-30	175	268	300

**Table 2 materials-15-05955-t002:** Elemental contents of ECNFs with different PA weight fractions.

	Atomic Concentration %			
	C 1s	N 1s	O 1s	P 2p
ENFs-0	90.12	4.06	5.82	-
ENFs-15	88.40	4.33	6.29	0.98
ENFs-30	88.36	4.69	5.97	0.98

**Table 3 materials-15-05955-t003:** Deconvolution results of XPS spectra of different PA weight fractions.

	ENFs-0		ENFs-15		ENFs-30	
	Position (eV)	At. (%)	Position (eV)	At. (%)	Position (eV)	At. (%)
**N 1s**	389.2	0.77	398.2	0.56	398.5	0.77
	400.7	1.94	400.9	2.85	400.8	3.16
	402.4	1.35	402.4	0.92	403.0	0.76
**P 2p**	-	-	129.6	0.07	129.6	0.17
	-	-	132.4	0.85	132.0	0.76
	-	-	134.2	0.06	136.0	0.05

**Table 4 materials-15-05955-t004:** The crystallinity structure and BET surface area of ECNFs with different PA weight fractions.

	2θ/°	d_002_/nm	L_c_/nm	L_c_/d_002_	S_BET_/m^2^·g^−1^
ENFs-0	25.1	0.3542	1.54	4.35	9.1
ENFs-15	24.4	0.3648	1.85	5.07	7.0
ENFs-30	24.6	0.3615	1.75	4.84	9.3

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
