# Peer review of "A Simple Method for Preparation of Highly Conductive Nitrogen/Phosphorus-Doped Carbon Nanofiber Films"

_materials, 2022, doi:10.3390/ma15175955_

Round 1

Reviewer 1 Report

1. The authors mention ORR in the title but in the abstract they don't discuss anything on ORR. I think they should add one or two lines about the results obtained in ORR in the abstract

2. In the abstract the authors should also mention the effect of the porosity i.e. it improved the ORR performance

3. The figures should be made more bigger so that they are clear for the readers

4. the authors should explain what Ti and TpC are in the manuscript

5. the authors should explain the effect of doping on the diameter or pore size. Does the porous nature of diameter size increase with increase in doping amount of PA? Why does it?

6. the authors should deconvolute the peaks of N and P and explain which configurations play a significant role in ORR, which ones act as active sites?

7. this manuscript states as per the title for ORR but I dont see efficient ORR data i.e. CV curves, LSV plots.

Reviewer 2 Report

The presented article "A Simple Method for Preparation of Highly Conductive Nitrogen/Phosphorus-Doped Carbon Nanofiber Films for the Oxygen Reduction Reaction" is devoted to obtaining new doped carbon fibers with high conductivity. An original method for the synthesis of the Highly Conductive Nitrogen/Phosphorus-Doped Carbon Nanofiber series was proposed, and the composition and structure of the obtained fibers were detail studied. However, before publishing the article, it is necessary to correct a number of remarks:

To study the obtained carbon nanofibers, it is necessary to measure the materials area by the BET method.

When analyzing the results of XPS, it is necessary to present the spectrum of nitrogen in more detail and analyze it, in what forms nitrogen is present in the resulting Nanofiber.

When analyzing the Raman spectra, it is necessary to explain the details of calculating the D/G ratio, namely, how the spectrum was fit.

In the title, the obtained carbon materials are presented for the Oxygen Reduction Reaction, while the activity of materials in ORR is not considered in the article itself. It is necessary to either add a section with the activity of materials in ORR to the article, or clarify the title.

According to the list of references, there are not enough recent works of 2020-2022.

Round 2

Reviewer 1 Report

The authors did a great job responding to the questions.

Reviewer 2 Report

The authors responded to the comments and made the necessary additions. The article can be accepted for publication.